psychology/neuroscience

reaction time, questionable research practices, *p*-hacking, false positives

**Author for correspondence:**
Luis Morís Fernández
e-mail: luis.moris.fernandez@gmail.com

# Flexibility in reaction time analysis: many roads to a false positive?

Luis Morís Fernández[1,2] and Miguel A. Vadillo[1]

[1]Departamento de Psicología Básica, Universidad Autónoma de Madrid, Madrid, Spain
[2]Multisensory Research Group, Center for Brain and Cognition, Universitat Pompeu Fabra, Barcelona, Spain

LM, 0000-0001-5247-7503; MAV, 0000-0001-8421-816X

In the present article, we explore the influence of undisclosed flexibility in the analysis of reaction times (RTs). RTs entail some degrees of freedom of their own, due to their skewed distribution, the potential presence of outliers and the availability of different methods to deal with these issues. Moreover, these degrees of freedom are usually not considered part of the analysis itself, but preprocessing steps that are contingent on data. We analysed the impact of these degrees of freedom on the false-positive rate using simulations over real and simulated data. When several preprocessing methods are used in combination, the false-positive rate can easily rise to 17%. This figure becomes more concerning if we consider that more degrees of freedom are awaiting down the analysis pipeline, potentially making the final false-positive rate much higher.

## 1. Introduction

John P. Hack is a cognitive scientist. He has just run an experiment comparing reaction times (RTs) in conditions A and B and he is pretty sure of his hypothesis that the median RT of the two conditions will differ, but when he analyses the data, he obtains a *p*-value of 0.07. Disappointed, John wonders what may have gone wrong. After checking some papers, he remembers that RT distributions can be quite skewed and sometimes medians are better than means. 'Maybe there are more outliers than expected in my data. I have to eliminate outliers more than two standards deviations away from the mean!', he determines. He opens his analysis script and changes one variable from 3 to 2. The *p*-value goes down to 0.052. Karen Transformáñez passes by the office and suggests he try the inverse transformation. 'I use it when things don't work with the logarithm', she says. John changes the name of a function in his script and, 'Eureka!', he exclaims, 'I knew the conditions were different' (*p*-value 0.047). Is his *p*-value

correct? No. Can he make any valid conclusion based on it? No. Did he have any malice when doing the above? No. Does John exist? Definitely yes and probably we are John ourselves.

Starting in the second half of the nineteenth century, Donders and Galton kick-started the use of RTs [1,2]. Since then, this measure has been used as a dependent variable in countless experiments. Nonetheless, the analysis of RTs is far from trivial. Firstly, their distribution is positively right-skewed, making it a less than perfect variable for classic statistical tests that require normal data (e.g. *t*-test or ANOVA). Secondly, RTs are inherently noisy, as they usually contain outliers that do not result from the process of interest [3].

Several alternatives exist to analyse RTs [4–6]. One of them is to ignore all these issues and just apply classical parametric statistics on raw data. Another possible solution is to transform the data (e.g. log-transform, inverse-transform or the square-root-transform) to better approximate a normal distribution. A third alternative is to remove outliers from the distribution by discarding observations above or a below a fixed cut-point, some number of standard deviations away from the mean, or a fixed percentage of the upper and lower observations. A fourth approach is to abandon central tendency measures and instead use alternative summaries of the data, such as Vincentizing, or rely on statistical models such as drift diffusion, ex-Gaussian or hazard models. Although the preceding list is not exhaustive, it serves to illustrate the following point: confronted with all these alternatives, which one is the most appropriate given one's study? Particularly, keeping in mind that many methods can be combined (e.g. calculating the mean for each participant and condition after log-transforming data, from which all trials below 50 ms and above 1000 ms have been removed, and then performing a *t*-test).

Recent studies [7–9] have shown that *researcher's degrees of freedom* can inflate the proportion of false positives. This increase takes place because researchers will typically only report the analyses that produced significant results and drop those that did not. In the particular case of RTs preprocessing, this problem can be aggravated for at least three reasons. Firstly, due to the right-skewed distribution of RTs and to the potential presence of outliers, many different preprocessing pipelines (PPs) exist, each one with different virtues and shortcomings, but all of them perfectly reasonable and valid *a priori*. Secondly, there is no single *ideal* pipeline for analysing RTs that will fit all paradigms and experiments. And, thirdly, because RT preprocessing takes place before statistical testing starts, a researcher may think that checking which preprocessing *works better* will have no impact on the proportion of false positives in subsequent tests.

Many papers [4–6,10–15] have addressed the problem of how to preprocess and analyse RTs. One potential issue is that, because these papers list many available options to preprocess RTs, they might leave the reader under the false impression that all of them can be tested sequentially until one of them 'works'. In the present article, we address the increase in the proportion of false positives caused by flexibility in the preprocessing of RTs by means of Monte Carlo simulations over two real and 12 simulated datasets. For each dataset, we performed 1 000 000 independent simulations, applying a range of preprocessing steps (i.e. different central measures, outlier removal criteria or data transformations). Then, we measured the percentage of simulations that obtained at least one statistically significant result at $\alpha = 0.05$ (two-tailed test) as a function of the number and type of preprocessing steps applied.

For the sake of simplicity, the statistical approach adopted in both cases relies on reducing the distribution of RTs to a single summary statistic, such as the mean or median. Although this approach has been challenged by alternative models, such as drift diffusion models or ballistic models that do not reduce the RT distribution to a single statistic, our simulations and analyses are based on summary statistics because much of the research based on RTs still follows this straightforward approach. Just as an example, in an informal sample of the methods adopted in recent studies, we found that 19 out of 25 sampled articles conducted an analysis of variance or *t*-test on average RTs (see the electronic supplementary material for more information). Note, however, that even drift diffusion and linear ballistic models require some data preprocessing, such as dealing with outliers [16,17], and, therefore, it is likely that the results of the present study still apply to them. Furthermore, these models may also have researchers' degrees of freedom or their own. It is not the goal of the present paper to find an optimal method to analyse RTs, but to alert the reader of the impact of flexibility in data preprocessing, regardless of the final test or model applied in statistical analyses. Readers interested in knowing more about the problems of each particular RT preprocessing method have an extensive literature available [4–6,10–15] at their disposal.

# 2. Method

## 2.1. Description of the datasets

In this study, we used two real datasets: one provided by Ebersole *et al.* [18], referred to as *Stroop* in this article, and another one provided by Rousselet & Wilcox [14], based on Ferrand *et al.* [19], referred to as *Flexicon* in this article. Both real datasets correspond to repeated measures designs comparing two conditions. In addition to using real data, we also generated simulated datasets from an ex-Gaussian distribution parametrized in 12 different ways.

### 2.1.1. Stroop

From the original dataset of Many Labs 3 [18], covering several experimental paradigms, we used only the data corresponding to the Stroop task. We refer the reader to the original paper for details on the task design. The full dataset contains 3348 participants with 63 trials each (see electronic supplementary material, figure S1). Before using these data in our simulations, we performed the following steps:

1. For each trial, only RT and participant ID were used.
2. After visualizing the RT distribution across participants, we found a small number of extremely long (greater than 10 s) or short (less than 1 ms) RTs. To trim the distribution, we calculated the 1 and 99 percentiles of all the trials and participants pooled together and dropped all trials with RTs below or above these thresholds, respectively (see electronic supplementary material, figure S1).
3. To avoid including participants contributing with very few trials, we dropped from the sample all participants retaining fewer than 50% of the original trials after applying step 2. Thirty-nine participants were dropped following this criterion (remaining $N = 3309$).
4. We equated the number of trials for all participants to 64, the closest even number, by resampling trials at random within each participant. This was done so that in each of the simulations described below, each participant contributed with the same number of trials and the dataset of each participant could be divided in two halves that would be later assigned to different conditions. This simplified the coding and speeded up the computations while making a negligible difference in the results.

### 2.1.2. Flexicon

The second dataset contained data from 959 participants with approximately 2000 trials each (see electronic supplementary material, figure S2). We refer the reader to the original paper [19] for details on the task design. The dataset was processed following the same steps as for the Stroop, with the only exception that in step 3, no participants were dropped and in step 4, we equated the number of trials to 2000.

### 2.1.3. Simulated data

We also generated simulated datasets sampling data from an ex-Gaussian distribution. The ex-Gaussian distribution has been shown to provide a good fit to empirical RT distributions [20–23] and has been used previously to simulate RT data [14,24]. This distribution is produced by the convolution of a Gaussian distribution and an exponential distribution and is defined by three parameters ($\mu$, $\sigma$, $\tau$). The first two refer to the mean and standard deviation of the Gaussian component, while the last one defines the exponential decay parameter. We created 12 different simulated datasets. Each dataset contained observations sampled from one of 12 ex-Gaussian distributions, each one with a different combination of parameters (table 1). This set of combinations of parameter values has been used in previous simulation studies [14,24] and is representative of RT values reported in empirical papers [21,22]. For each simulated dataset, we generated 1 000 000 observations that were randomly divided in 10 000 subsets, each one representing a hypothetical participant, of 100 observations each. These simulated datasets allowed us to test the generalizability of the results obtained with the two real datasets, as they cover a wider range of possible distributions of RTs.

## 2.2. Selection of preprocessing pipelines and simulation

The goal of this study was to determine the proportion of false positives obtained when several PPs are applied to a dataset and only the one with the lowest *p*-value is reported. We first describe the subset of PPs we considered in this study and the algorithm used for the simulations.

**Table 1.** Parameters sets for simulated datasets.

| $\mu$ | 300 | 300 | 350 | 350 | 400 | 400 | 450 | 450 | 500 | 500 | 550 | 550 |
|---|---|---|---|---|---|---|---|---|---|---|---|---|
| $\sigma$ | 20 | 50 | 20 | 50 | 20 | 50 | 20 | 50 | 20 | 50 | 20 | 50 |
| $\tau$ | 300 | 300 | 250 | 250 | 200 | 200 | 150 | 150 | 100 | 100 | 50 | 50 |

The pipelines used in the following simulations differed in terms of (i) central tendency measure, (ii) threshold values for removing outliers, and (iii) transformations of data. We selected the mean and median as central tendency measures, as they are very often used in RT analyses, despite the controversy about their use [14,24]. Threshold values for removing outliers were set to 2 s.d., 2.5 s.d. or 3 s.d., based on a survey of the literature conducted by Leys *et al.* [10]. Two transformations (logarithm and inverse) were selected among the ones typically used in the RT literature [5] and we also included the possibility of not transforming the data.

We ran 1 000 000 iterations for each of the real and simulated datasets. In each iteration, we created a random sample of participants, assigned the trials randomly to either of two conditions, applied the previously described PPs and ran a *t*-test for each of those PPs:

1. *Participants' sample.* Thirty participants were sampled at random from the dataset. For each participant, all trials were selected (i.e. 64, 2000 or 100, depending on the dataset). We chose an *N* of 30 because it is a reasonable number of participants for a hypothetical study and because it allowed resampling with little overlap across simulations.
2. *Force a null effect.* To force a null effect on each iteration, we created two conditions artificially by randomly assigning one half of the trials to each condition, regardless of their original assignment in the case of the real datasets [25].
3. *Preprocessing and summary statistic.* For each participant, we applied different combinations of PPs. Firstly, data were either not transformed (Raw), log-transformed (Log) or inverse-transformed (Inv). Secondly, outliers were either not excluded or excluded using as cut-off values either 2 s.d., 2.5 s.d. or 3 s.d. away from the participants mean (no s.d., 2 s.d., 2.5 s.d. and 3 s.d., respectively). Finally, for each participant, condition and preprocessing combination described before, we calculated the mean RT (Mean) and the median RT (Median). This means that in our simulations RTs could be submitted to 24 (3 transformations × 4 outlier exclusion methods × 2 summary statistics) different PPs. In the following sections, PPs will be named after their summary statistic, transformation and outlier detection cut-off (e.g. Mean–Log–2 s.d.).
4. *Statistical test.* Finally, we ran a two-tailed paired *t*-test comparing the two conditions separately for each PP (figure 1).

## 2.3. Analysis

We estimated the proportion of false positives under different sets of PPs by calculating the proportion of iterations in which we found a significant *p*-value ($\alpha = 0.05$) in at least one of the PPs considered. For example, imagine that we consider the set of two PPs {Median–Raw–3 s.d., Mean–Raw–3 s.d.}. We would use the proportion of iterations in which we found a significant *p*-value in the first, in the second or in both PPs as an estimator of the proportion of false positives.

## 3. Results

Table 2 and figure 2 show the estimated proportion of false positives for different sets of PPs. For the sake of simplicity, only a representative subset of all possible combinations ($2^{24}$ in total) are shown in table 2. The proportion of false positives was equal to the $\alpha$-level of the *t*-test, 0.05, only when a single PP was applied. As the size of the PP set grows, the proportion of false positives increases progressively in all three datasets.

## 4. Discussion

RT data are difficult to analyse due to their right-skewed distribution and the potential presence of outliers. Several techniques exist to address both problems. From the myriad of possible approaches, we selected a subset of methods and analysed the increase of false positives produced by the

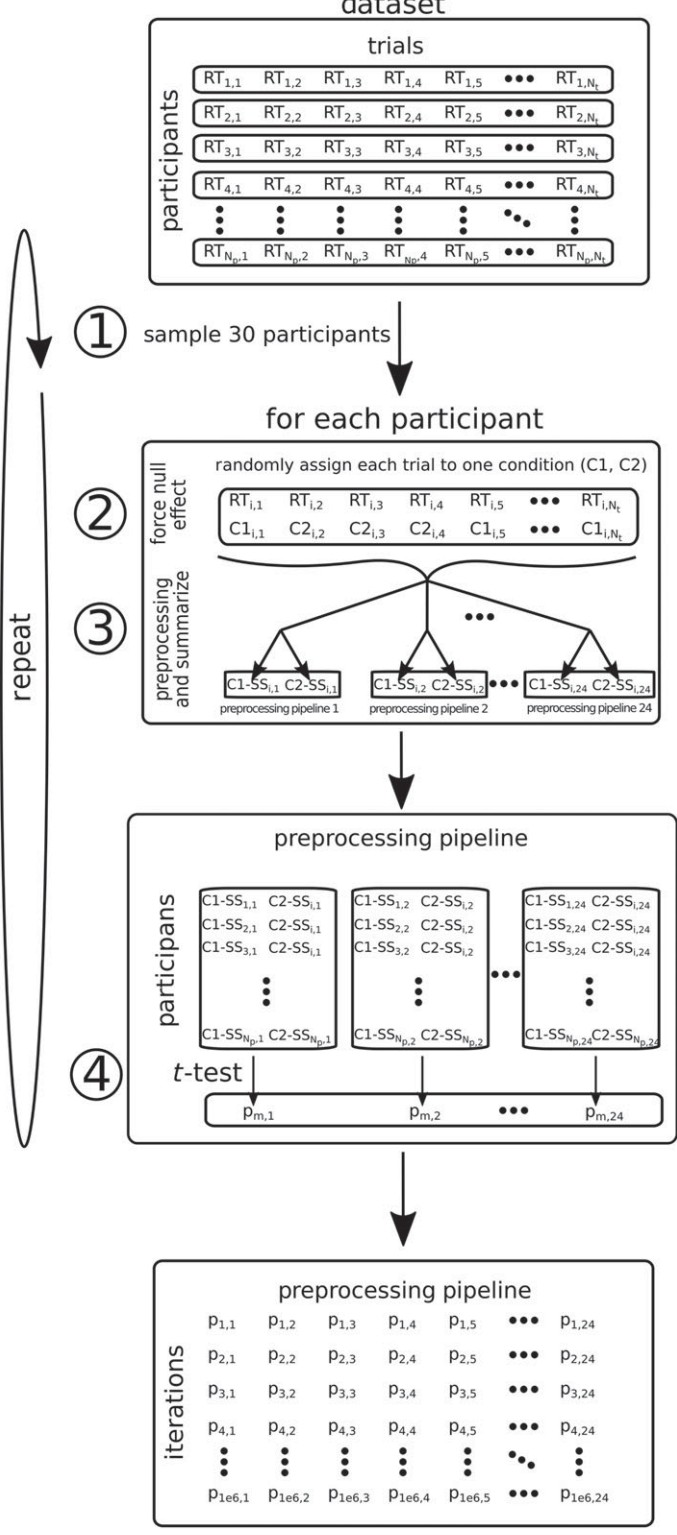

**Figure 1.** Schema of the simulation. Circled numbers correspond to the steps described in the main text. $RT_{i,j}$ represents the RT of participant $i$ in trial $j$. $C1_{i,j}$ and $C2_{i,j}$ denote conditions 1 or 2, respectively, assigned to each $RT_{i,j}$ in a given iteration. $C1\text{-}SS_{i,k}$ and $C2\text{-}SS_{i,k}$ represent the summary statistic in a given iteration of conditions 1 and 2, respectively, for participant $i$ after applying the $k$th PP of the 24 applied in the simulation. $p_{m,k}$ represent the $p$-value when a two-tailed paired $t$-test is performed between conditions 1 and 2 in the iteration $m$ when the $k$th PP is applied.

application of several of them in turn. Our results indicate that the seemingly innocuous decision of using either the mean or the median as a summary statistic can increase the proportion of false positives from 0.05 to 0.08, or up to 0.09, if several outlier exclusion techniques are used. The false-positive rate can

**Table 2.** Proportion of significant tests depending on the PP set.

| set of PP | real | | simulated | | | | | | | | | | | |
|---|---|---|---|---|---|---|---|---|---|---|---|---|---|---|
| | | | μ | 300 | 300 | 350 | 350 | 400 | 400 | 450 | 450 | 500 | 500 | 550 | 550 |
| | | | σ | 20 | 50 | 20 | 50 | 20 | 50 | 20 | 50 | 20 | 50 | 20 | 50 |
| | | | τ | 300 | 300 | 250 | 250 | 200 | 200 | 150 | 150 | 100 | 100 | 50 | 50 |
| | Flexicon | Stroop | | | | | | | | | | | | | |
| Median–Raw–No s.d. | 0.06 | 0.06 | | 0.05 | 0.05 | 0.05 | 0.05 | 0.05 | 0.05 | 0.05 | 0.05 | 0.05 | 0.05 | 0.05 | 0.05 |
| Median–Log–No s.d. | | | | | | | | | | | | | | | |
| Median–Inverse–No s.d. | | | | | | | | | | | | | | | |
| Mean–Raw–No s.d. | 0.07 | 0.07 | | 0.07 | 0.08 | 0.07 | 0.07 | 0.07 | 0.07 | 0.07 | 0.07 | 0.07 | 0.07 | 0.07 | 0.07 |
| Mean–Log–No s.d. | | | | | | | | | | | | | | | |
| Mean–Inverse–No s.d. | | | | | | | | | | | | | | | |
| Median–Raw–No s.d. | 0.08 | 0.08 | | 0.08 | 0.08 | 0.08 | 0.08 | 0.08 | 0.08 | 0.08 | 0.08 | 0.08 | 0.08 | 0.08 | 0.08 |
| Mean–Raw–No s.d. | | | | | | | | | | | | | | | |
| Median–Raw–No s.d. | 0.10 | 0.11 | | 0.10 | 0.10 | 0.10 | 0.10 | 0.10 | 0.10 | 0.10 | 0.10 | 0.10 | 0.10 | 0.10 | 0.10 |
| Median–Log–No s.d. | | | | | | | | | | | | | | | |
| Median–Inverse–No s.d. | | | | | | | | | | | | | | | |
| Mean–Raw–No s.d. | | | | | | | | | | | | | | | |
| Mean–Log–No s.d. | | | | | | | | | | | | | | | |
| Mean–Inverse–No s.d. | | | | | | | | | | | | | | | |
| Mean–Raw–2 s.d. | 0.08 | 0.08 | | 0.09 | 0.09 | 0.09 | 0.09 | 0.09 | 0.09 | 0.09 | 0.09 | 0.09 | 0.09 | 0.09 | 0.09 |
| Mean–Raw–2.5 s.d. | | | | | | | | | | | | | | | |
| Mean–Raw–3 s.d. | | | | | | | | | | | | | | | |
| all (24 PP) | 0.16 | 0.17 | | 0.16 | 0.16 | 0.16 | 0.16 | 0.16 | 0.16 | 0.16 | 0.16 | 0.16 | 0.16 | 0.16 | 0.16 |

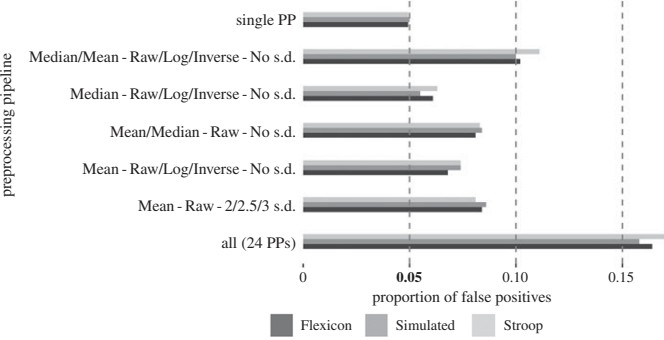

**Figure 2.** Proportion of false positives as a function of the PP set. Each group of bars represents the proportion of false positives obtained in the simulation for each dataset, given a specific combination of PPs. Each group name indicates which PPs were applied, slash bars separate variations of the same parameter included in a PP (e.g. Median–Raw/Log/Inverse–No s.d. indicates that Median–Raw–No s.d., Median–Log–No s.d. and Median–Inverse–No s.d. pipelines are considered). The proportion of false positives was equal to the $\alpha$-level of the $t$-test, 0.05, only when a single PP was applied.

increase beyond 0.15 if all the described PPs are used (see third, fifth and last rows of table 2). These numbers become particularly alarming if we consider additional degrees of freedom [9] that can increase the false-positive rate even further. In this sense, our results are probably an underestimation of the actual false-positive rate, as we have not exhaustively covered all possible PPs. It could be argued that most researchers would never try all possible pipelines and combinations, and while we partially agree, we suspect that many researchers working with RTs are likely to deem the exploration of several approaches as not only valid, but even necessary to properly analyse RT data. Even more, it is not necessary to perform all possible PPs every time. Researchers probably stop trying PPs once they find one that provides a significant result. Also, they probably do not explore PPs randomly but in an intelligent way by, for example, selecting the transformation that maximizes the difference between conditions. Note that whether or not a researcher actually explores all PPs or not is irrelevant: just considering them or being willing to use them is enough to inflate the false-positive rate [7].

Preregistration provides effective means to avoid these problems. The main idea behind preregistration consists of declaring *a priori* the specific procedures and the data analysis approach, including any preprocessing of RTs. In a preregistered study, all the researcher's degrees of freedom are squeezed into a predefined preprocessing and analysis pipeline [9,26,27], drawing a sharp distinction between confirmatory and exploratory analyses [28,29]. Different approaches to deal with distributional assumptions in preregistered research have been proposed [30].

We must highlight that flexibility in data preprocessing is not the only factor with an impact on the false-positive rate in analyses of RTs. For example, differences in skewness between conditions can also raise the number of false positives above 5% [14] (note that no difference in skewness was present in the simulations reported in this study as our interest was to detect false positives when distributions in both conditions were identical). Therefore, researchers must be aware of other possible sources of false positives even when using only a single preregistered analysis pipeline.

Given that the flexibility in the preprocessing of RTs is largely driven by attempts to correct for the marked skewness of their distribution, the problems highlighted in this article could be ameliorated by resorting to analytic methods specifically designed to deal with non-normal distributions, such as bootstrapping or permutations tests, possibly combined with robust statistics such as the trimmed mean [31]. This being said, relying on bootstrapping or similar methods does not preclude the need to preregister studies, as the inflation of the false-positive rate discussed here is not produced by the use of one statistical approach or another, but by selecting among different analytic approaches based on significance.

The present study is not without limitations. Firstly, we did not test all possible PPs because the number of potential combinations grows exponentially with each additional PP and so does computational time. A second shortcoming is the limited availability of real datasets to perform this kind of simulations. Although smaller datasets are available, only large datasets with information at the trial level are suitable for the present simulations. All the scripts and data used in the present study are available online [32] and, therefore, all simulations can be easily repeated with different datasets, PPs or statistical tests.

Data accessibility. All materials, data and scripts used for this article are publicly available at: https://osf.io/q3d4m/.
Authors' contributions. L.M.F. and M.A.V. developed the study concept. L.M.F. performed the simulations and the analysis. L.M.F. drafted the manuscript. L.M.F. and M.A.V. reviewed the manuscript. Both authors approved the final version of the manuscript for submission.
Competing interests. The authors declare that they have no competing interests.
Funding. The present work was supported by grant no. 2016-T1/SOC-1395 from Comunidad de Madrid (Programa de Atracción de Talento Investigador), grant nos. PSI2017-85159-P and PSI2016-75558-P from Agencia Estatal de Investigación (AEI, National Research Agency) and Fondo Europeo de Desarrollo Regional (FEDER, European Regional Development Fund), and grant no. 2017 SGR 1545 financed by the AGAUR (Catalan government).

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
