## [Reviewer comments · Royal Society Open Science]

Review History

RSOS-190831.R0 (Original submission)

Review form: Reviewer 1 (Guillaume Rousselet)

Is the manuscript scientifically sound in its present form?

No

Are the interpretations and conclusions justified by the results?

No

Is the language acceptable?

Yes

Do you have any ethical concerns with this paper?

No

Have you any concerns about statistical analyses in this paper?

Yes

Recommendation?

Major revision is needed (please make suggestions in comments)

Comments to the Author(s)

The paper presents a useful contribution but it is hard to read, contains misleading simplifications and a few major problems. The main improvements needed are:

- new ex-Gaussian simulations;
- clearer descriptions of the simulations (I don't think a non-specialist will understand them);
- new simulations for the permutation test;
- extension of some of the topics, so that readers don't leave with somewhat misleading information.

Introduction

"other common variables that follow a normal distribution" - I think you will find it difficult to list even a few such variables in neuroscience and psychology. In these fields, most continuous distributions are skewed, and some common distributions are never normal: percent correct data are beta-binomial, Likert scale scores are ordinal...

"anomalous distribution of RTs" - I don't understand what's anomalous with RTs. Skewed is the way they are. What's anomalous is the behaviour of users who apply tools that make unmatched assumptions about the data. I would encourage you to try to be more careful about the phrasing of many of the descriptions.

"ubiquitous presence of outliers" - how do you know that? It's very difficult to tease apart true outliers from genuine samples from skewed distributions, particularly with small sample sizes or heavy tails. Saying that outliers are likely would be more cautious.

You should mention that cut-offs and SD rules can introduce bias. Also, SD rules are not robust.

Study 1

Explain why you didn't include cut-offs. Feels like an omission after reading the introduction.

What affects the type I error rate is the difference in skewness between groups, so all the ex-Gaussian simulations are redundant: you could simply use a normal distribution and would get the same results. Instead, you need to keep one distribution constant (normal), and increase the skewness of the other distribution.

"For each of these simulated datasets we sampled 1,000,000 observations that were arbitrarily divided in 10,000 participants with 100 observations each." - so that way you created 12 populations from which you sampled in the simulations?

Figure 1: why is there a vertical white bar at 0.05 increments?

"The expected .05 proportion of false positives was obtained only in the ideal case in which a single PP was applied" - this should be illustrated too. Also, the expected 0.05 won't be obtained even with a single PP in the new ex-Gaussian simulations (see below).

Study 2

I do not understand the rationale for study 2. A permutation test is not the same as a t-test, they do not ask the same questions about the data. In particular, the permutation is done at the trial level, so the inference is not the same as the t-test inference. We know there are methods with higher power than t-tests on means, and it certainly doesn't hurt to remind users of that, but it is misleading to present a rather unusual permutation test as the solution. For comparison with the t-test, you also need to report the performance of the permutation test in Study 1. Indeed, it wouldn't be appropriate to recommend a method based on power without checking false positives.

Why not include other parametric and non-parametric tests as well? IF the permutation is just used as a demo, then that's fine, but please inform readers about other well-documented options. For instance, to protect against the influence of skewness and outliers, I don't think it is a good idea to recommend users to make inferences about the mean using a permutation test. Trimmed means would be a much better recommendation, and they can be combined with a parametric test, permutation, and at least two flavours of bootstrap (see work by Rand Wilcox on the topic).

For completeness, I would suggest to look at power using the synthetic ex-Gaussian data, with varying skewness. This should help tease apart the different techniques.

Step 1: is the proportion of trials per condition?

"The difference in means between the two artificial conditions was calculated and added to the null distribution" - do you mean the value was saved, so at the end you had a distribution of 1,000 differences under the assumption of exchangeability of trials (not participants)?

"we also included in the null distribution the real observed difference between means corresponding to the original" - alternatively, and perhaps easier to explain, you could simply replace 0 p values with 1/1000.

Discussion

The proportion of false positives is only expected to be at 0.05 if the two distributions compared have the same skewness. When distributions differ in skewness, the number of false positives increases. In the presence of outliers, it goes down. So it is misleading to describe an expected 0.05 nominal level when using only one preprocessing pipeline.

Using the mean or the median addresses different questions about the data, and both have advantages and limitations in terms of power and bias depending on the shape of the populations. This should be mentioned, so that readers are not under the impression that the mean and the median are interchangeable.

Decision letter (RSOS-190831.R0)

28-Sep-2019

Dear Dr Morís Fernández,

The editors assigned to your paper ("Reaction times: Many ways of inadvertently obtaining a false positive") have now received comments from reviewers. We would like you to revise your paper in accordance with the referee and Associate Editor suggestions which can be found below (not including confidential reports to the Editor). Please note this decision does not guarantee eventual acceptance.

Please submit a copy of your revised paper before 21-Oct-2019. Please note that the revision deadline will expire at 00.00am on this date. If we do not hear from you within this time then it will be assumed that the paper has been withdrawn. In exceptional circumstances, extensions may be possible if agreed with the Editorial Office in advance. We do not allow multiple rounds of revision so we urge you to make every effort to fully address all of the comments at this stage. If deemed necessary by the Editors, your manuscript will be sent back to one or more of the original reviewers for assessment. If the original reviewers are not available, we may invite new reviewers.

- Data accessibility

If you wish to submit your supporting data or code to Dryad (<http://datadryad.org/>), or modify your current submission to dryad, please use the following link:
<http://datadryad.org/submit?journalID=RSOS&manu=RSOS-190831>

- Competing interests

- Authors' contributions

AB carried out the molecular lab work, participated in data analysis, carried out sequence alignments, participated in the design of the study and drafted the manuscript; CD carried out the statistical analyses; EF collected field data; GH conceived of the study, designed the study,

coordinated the study and helped draft the manuscript. All authors gave final approval for publication.

- Acknowledgements

- Funding statement

on behalf of Dr Narayanan Srinivasan (Associate Editor) and Essi Viding (Subject Editor)
openscience@royalsociety.org

Associate Editor's comments (Dr Narayanan Srinivasan):

Associate Editor: 1

Comments to the Author:

One expert reviewer has now commented on the paper. The comments are fairly extensive and i concur with most of them. Authors are requested to respond in detail for all the comments if and when they submit their next revision.

Comments to Author:

Reviewers' Comments to Author:

Reviewer: 1

Comments to the Author(s)

The paper presents a useful contribution but it is hard to read, contains misleading simplifications and a few major problems. The main improvements needed are:

- new ex-Gaussian simulations;
- clearer descriptions of the simulations (I don't think a non-specialist will understand them);
- new simulations for the permutation test;
- extension of some of the topics, so that readers don't leave with somewhat misleading information.

Introduction

"other common variables that follow a normal distribution" - I think you will find it difficult to list even a few such variables in neuroscience and psychology. In these fields, most continuous distributions are skewed, and some common distributions are never normal: percent correct data are beta-binomial, Likert scale scores are ordinal...

"anomalous distribution of RTs" - I don't understand what's anomalous with RTs. Skewed is the way they are. What's anomalous is the behaviour of users who apply tools that make unmatched assumptions about the data. I would encourage you to try to be more careful about the phrasing of many of the descriptions.

“ubiquitous presence of outliers” - how do you know that? It’s very difficult to tease apart true outliers from genuine samples from skewed distributions, particularly with small sample sizes or heavy tails. Saying that outliers are likely would be more cautious.

You should mention that cut-offs and SD rules can introduce bias. Also, SD rules are not robust.

Study 1

Explain why you didn’t include cut-offs. Feels like an omission after reading the introduction.

What affects the type I error rate is the difference in skewness between groups, so all the ex-Gaussian simulations are redundant: you could simply use a normal distribution and would get the same results. Instead, you need to keep one distribution constant (normal), and increase the skewness of the other distribution.

“For each of these simulated datasets we sampled 1,000,000 observations that were arbitrarily divided in 10,000 participants with 100 observations each.” - so that way you created 12 populations from which you sampled in the simulations?

Figure 1: why is there a vertical white bar at 0.05 increments?

“The expected .05 proportion of false positives was obtained only in the ideal case in which a single PP was applied” - this should be illustrated too. Also, the expected 0.05 won’t be obtained even with a single PP in the new ex-Gaussian simulations (see below).

Study 2

I do not understand the rationale for study 2. A permutation test is not the same as a t-test, they do not ask the same questions about the data. In particular, the permutation is done at the trial level, so the inference is not the same as the t-test inference. We know there are methods with higher power than t-tests on means, and it certainly doesn’t hurt to remind users of that, but it is misleading to present a rather unusual permutation test as the solution. For comparison with the t-test, you also need to report the performance of the permutation test in Study 1. Indeed, it wouldn’t be appropriate to recommend a method based on power without checking false positives.

Why not include other parametric and non-parametric tests as well? IF the permutation is just used as a demo, then that’s fine, but please inform readers about other well-documented options. For instance, to protect against the influence of skewness and outliers, I don’t think it is a good idea to recommend users to make inferences about the mean using a permutation test. Trimmed means would be a much better recommendation, and they can be combined with a parametric test, permutation, and at least two flavours of bootstrap (see work by Rand Wilcox on the topic).

For completeness, I would suggest to look at power using the synthetic ex-Gaussian data, with varying skewness. This should help tease apart the different techniques.

Step 1: is the proportion of trials per condition?

“The difference in means between the two artificial conditions was calculated and added to the null distribution” - do you mean the value was saved, so at the end you had a distribution of 1,000 differences under the assumption of exchangeability of trials (not participants)?

“we also included in the null distribution the real observed difference between means corresponding to the original” - alternatively, and perhaps easier to explain, you could simply replace 0 p values with 1/1000.

Discussion

The proportion of false positives is only expected to be at 0.05 if the two distributions compared have the same skewness. When distributions differ in skewness, the number of false positives increases. In the presence of outliers, it goes down. So it is misleading to describe an expected 0.05 nominal level when using only one preprocessing pipeline.

Using the mean or the median addresses different questions about the data, and both have advantages and limitations in terms of power and bias depending on the shape of the populations. This should be mentioned, so that readers are not under the impression that the mean and the median are interchangeable.

Author's Response to Decision Letter for (RSOS-190831.R0)

See Appendix A.

RSOS-190831.R1 (Revision)

Review form: Reviewer 1 (Guillaume Rousselet)

Is the manuscript scientifically sound in its present form?

Yes

Are the interpretations and conclusions justified by the results?

Yes

Is the language acceptable?

No

Do you have any ethical concerns with this paper?

Yes

Have you any concerns about statistical analyses in this paper?

No

Recommendation?

Accept with minor revision (please list in comments)

Comments to the Author(s)

The authors have addressed my comments and significantly improved the article. I have only minor comments and suggestions at this stage. I don't think the permutation test results should be included, even as supplementary information, unless the simulations are done at the participant level, as in study 1, and false positives are assessed too. Otherwise some readers might think it is ok to combine trials across participants, a practice that alters degrees of freedom and tests a very different hypothesis than what researchers typically have in mind:

<https://psyarxiv.com/jqw35>

<https://elifesciences.org/articles/48175>

I still do not understand how many simulation iterations were performed. I tend to think of a simulation iteration as a unit in which participants and trials are sampled, tests are applied and results recorded. I would not describe a simulated observation as a simulation. And I do not think this is typically the case - e.g.

<https://onlinelibrary.wiley.com/doi/full/10.1002/sim.8086>

So in your study, the most important information is the number of values used to compute the proportion of false positives. From your code, it seems that you performed 10,000 simulations. Is that correct?

On the OSF, I couldn't find a table describing all the files and folders. I suggest that you add a README file explaining the content of the different folders, and how to re-use the code to reproduce the simulations and investigate the results.

Sentence two of the abstract needs to be rephrased to avoid repeating "due to" and to improve clarity:

"RTs entail some degrees of freedom of their own, due to the several techniques present due to their non-normal skewed distribution and the potential presence of outliers."

Page 6: "RT modelling may also have researchers' degrees of freedom" - this is misleading because it suggests that a t-tests and ANOVAs are not models.

Page 9: "a revision of the literature" should be "a survey"?

Page 13: in the title of figure 2, I would spell out PP for clarity.

Page 18: "due to the RTs distribution" - this needs rephrasing

Page 19: "please, that..." -> "please note that..."?

Decision letter (RSOS-190831.R1)

12-Dec-2019

Dear Dr Morís Fernández:

On behalf of the Editors, I am pleased to inform you that your Manuscript RSOS-190831.R1 entitled "Flexibility in reaction time analysis: Many roads to a false positive?" has been accepted for publication in Royal Society Open Science subject to minor revision in accordance with the referee suggestions. Please find the referees' comments at the end of this email.

The reviewers and Subject Editor have recommended publication, but also suggest some minor revisions to your manuscript. Therefore, I invite you to respond to the comments and revise your manuscript.

- Ethics statement

- Data accessibility

<http://datadryad.org/submit?journalID=RSOS&manu=RSOS-190831.R1>

- Competing interests

- Authors' contributions

- Acknowledgements

- Funding statement

Because the schedule for publication is very tight, it is a condition of publication that you submit the revised version of your manuscript before 21-Dec-2019. Please note that the revision deadline will expire at 00.00am on this date. If you do not think you will be able to meet this date please let me know immediately.

To revise your manuscript, log into <https://mc.manuscriptcentral.com/rsos> and enter your Author Centre, where you will find your manuscript title listed under "Manuscripts with Decisions". Under "Actions," click on "Create a Revision." You will be unable to make your

revisions on the originally submitted version of the manuscript. Instead, revise your manuscript and upload a new version through your Author Centre.

Kind regards,
Anita Kristiansen
Editorial Coordinator
Royal Society Open Science
openscience@royalsociety.org

on behalf of Dr Narayanan Srinivasan (Associate Editor) and Essi Viding (Subject Editor)
openscience@royalsociety.org

Associate Editor Comments to Author (Dr Narayanan Srinivasan):

The manuscript is much improved and reviewer 1 is mostly satisfied with the revision. However, R1 still has some concerns and I request the authors to address these concerns and submit the final version.

Reviewer comments to Author:

Reviewer: 1

Comments to the Author(s)

The authors have addressed my comments and significantly improved the article. I have only minor comments and suggestions at this stage. I don't think the permutation test results should be included, even as supplementary information, unless the simulations are done at the participant level, as in study 1, and false positives are assessed too. Otherwise some readers might think it is ok to combine trials across participants, a practice that alters degrees of freedom and tests a very different hypothesis than what researchers typically have in mind:

<https://psyarxiv.com/jqw35>

<https://elifesciences.org/articles/48175>

I still do not understand how many simulation iterations were performed. I tend to think of a simulation iteration as a unit in which participants and trials are sampled, tests are applied and results recorded. I would not describe a simulated observation as a simulation. And I do not think this is typically the case - e.g.

<https://onlinelibrary.wiley.com/doi/full/10.1002/sim.8086>

So in your study, the most important information is the number of values used to compute the proportion of false positives. From your code, it seems that you performed 10,000 simulations. Is that correct?

On the OSF, I couldn't find a table describing all the files and folders. I suggest that you add a README file explaining the content of the different folders, and how to re-use the code to reproduce the simulations and investigate the results.

Sentence two of the abstract needs to be rephrased to avoid repeating "due to" and to improve clarity:

"RTs entail some degrees of freedom of their own, due to the several techniques present due to their non-normal skewed distribution and the potential presence of outliers."

Page 6: "RT modelling may also have researchers' degrees of freedom" - this is misleading because it suggests that a t-tests and ANOVAs are not models.

Page 9: "a revision of the literature" should be "a survey"?

Page 13: in the title of figure 2, I would spell out PP for clarity.

Page 18: "due to the RTs distribution" - this needs rephrasing

Page 19: "please, that..." -> "please note that..."?

Author's Response to Decision Letter for (RSOS-190831.R1)

See Appendix B.

Decision letter (RSOS-190831.R2)

17-Jan-2020

Dear Dr Morís Fernández,

It is a pleasure to accept your manuscript entitled "Flexibility in reaction time analysis: Many roads to a false positive?" in its current form for publication in Royal Society Open Science. The comments of the reviewer(s) who reviewed your manuscript are included at the foot of this letter.

on behalf of Dr Narayanan Srinivasan (Associate Editor) and Essi Viding (Subject Editor)
openscience@royalsociety.org

Appendix A

	Comments made by Reviewer 1	Changes to the manuscript
	The paper presents a useful contribution but it is hard to read, contains misleading simplifications and a few major problems. The main improvements needed are: - new ex-Gaussian simulations;	Please, find our responses below to specific concerns about the current simulations.
	- clearer descriptions of the simulations (I don't think a non-specialist will understand them);	Following R1's advice, we have included a flow chart summarizing the simulation procedure that, we hope, will make the simulations clearer to the readers.
	- new simulations for the permutation test;	For the reasons explained below, we have decided to remove Study 2 from the ms.
	- extension of some of the topics, so that readers don't leave with somewhat misleading information.	As explained above and in the following sections, we have made specific changes addressing all the topics mentioned by R1.
Q1	## Introduction "other common variables that follow a normal distribution" - I think you will find it difficult to list even a few such variables in neuroscience and psychology. In these fields, most continuous distributions are skewed, and some common distributions are never normal: percent correct data are beta-binomial, Likert scale scores are ordinal...	We agree with R1 in this point and have edited the ms consequently at several places.
Q2	"anomalous distribution of RTs" - I don't understand what's anomalous with RTs. Skewed is the way they are. What's anomalous is the behaviour of users who apply tools that make unmatched assumptions about the data. I would encourage you to try to be more careful about the phrasing of many of the descriptions.	Again, we agree that is incorrect to refer to the distribution of RTs as "anomalous" and have edited the text to avoid this expression throughout the ms.
Q3	"ubiquitous presence of outliers" - how do you know that? It's very difficult to tease apart true outliers from genuine samples from skewed distributions, particularly with small sample sizes or heavy tails. Saying that outliers are likely would be more cautious.	Following R1's advice we have rewritten this sentence.
Q4	You should mention that cut-offs and SD rules can introduce bias. Also, SD rules are not robust.	We agree with R1 in that cut-offs and SD rules can introduce bias and are not robust. We have pointed out that some methods mentioned in this paper have their own problems and we direct the reader to the available literature for the possible issues regarding each different RT preprocessing technique. For instance, at the end of the introduction we now include the following text: "It is not the goal of the present paper

		to find an optimal method to analyse RTs, but to alert the reader of the impact of flexibility in data preprocessing, regardless of the final test or model applied in statistical analyses. Readers interested in knowing more about the problems of each particular RT preprocessing method have an extensive literature available [4–6,10–15] at their disposal.”
Q5	## Study 1 Explain why you didn't include cut-offs. Feels like an omission after reading the introduction.	We did include cut-offs in our simulations in Study 1. This was explicitly mentioned in the main text under the epigraph Study 1: “The pipelines used in the following simulations differed in terms of (a) central tendency measure, (b) threshold values for removing outliers, and (c) transformations of data. We selected the mean and median as central tendency measures, as they are very often used in RT analyses, despite the controversy about their use ... Threshold values for removing outliers were set to 2, 2.5 or 3 standard deviations, based on a revision of the literature conducted by Leys et al.”
Q6	What affects the type I error rate is the difference in skewness between groups, so all the ex-Gaussian simulations are redundant: you could simply use a normal distribution and would get the same results. Instead, you need to keep one distribution constant (normal), and increase the skewness of the other distribution.	Before we reply to this comment in detail, we would like to highlight that our paper does not try to cover all possible factors that could influence Type I error rate, but just those that are related to flexibility in data analysis. This being said, we agree that differences in skewness across conditions can also increase the error rate. Table R1, at the end of the present file, shows the results of several simulations where we have manipulated the degree of skewness across conditions but keeping the mean of the Ex-Gaussian distribution constant (note that the mean of the distribution is given by $\mu + \tau$). As indicated by R1, it is indeed true that the false-positive rate is slightly higher than .05 (specifically, between .051 and .052). See also response to Q14.
Q7	“For each of these simulated datasets we sampled 1,000,000 observations that were arbitrarily divided in 10,000 participants with 100 observations each.” - so that way you created 12 populations from which you sampled in the simulations?	We now clarify in the main text: “We created 12 different simulated datasets. Each dataset contained observations sampled from one of 12 ex-Gaussian distributions, each one with a different combination of parameters (see Table I). This set of combinations of parameter values has been used in previous simulation

		studies [14,24] and is representative of RT values reported in empirical papers [21,22]. For each simulated dataset, we generated 1,000,000 observations that were randomly divided in 10,000 subsets, each one representing a hypothetical participant, of 100 observations each. These simulated datasets allowed us to test the generalizability of the results obtained with the two real datasets, as they cover a wider range of possible distributions of RTs."
Q8	Figure 1: why is there a vertical white bar at 0.05 increments?	The white bars represented grid lines. We have modified the figure to improve the readability of the graph.
Q9	"The expected .05 proportion of false positives was obtained only in the ideal case in which a single PP was applied" - this should be illustrated too. Also, the expected 0.05 won't be obtained even with a single PP in the new ex-Gaussian simulations (see below).	To illustrate this point, we have included an additional group in the bar graph (Figure 2) corresponding to the average false-positive rate when a single PP is used. Our simulations do not indicate an increase in false positives when a single PP is used and both distributions are identical (see Table RII at the end of the present document and Figure 2 in the main text). As shown in Table RI, the slight increase in the false positive rate only occurs when comparing distributions with different skewness, a situation that is beyond the scope of the present paper. See also response to Q14.
Q10	## Study 2 I do not understand the rationale for study 2. A permutation test is not the same as a t-test, they do not ask the same questions about the data. In particular, the permutation is done at the trial level, so the inference is not the same as the t-test inference. We know there are methods with higher power than t-tests on means, and it certainly doesn't hurt to remind users of that, but it is misleading to present a rather unusual permutation test as the solution. For comparison with the t-test, you also need to report the performance of the permutation test in Study 1. Indeed, it wouldn't be appropriate to recommend a method based on power without checking false positives. Why not include other parametric and non-parametric tests as well? If the permutation is just used as a demo, then that's fine, but please inform readers about other well-documented options. For instance, to protect against the influence of skewness and outliers, I don't think it is a good idea to recommend users to make inferences about the mean using a permutation test. Trimmed means would be a much better recommendation, and they can be combined with a parametric test, permutation, and at least two flavours of bootstrap (see work by Rand Wilcox on the topic). For completeness, I would suggest to look at power using the synthetic ex-Gaussian data, with varying skewness. This should help tease apart the different techniques.	After carefully considering R1's concerns, we have decided to remove Study 2 from the present manuscript. Rousselet, Pernet, & Wilcox (2019) and Rousselet & Wilcox (2018) highlight the problems of using the mean with bootstrapping techniques and, consistent with R1, they also recommend the use of trimmed means. We think that removing Study 2 will improve the clarity of the ms and highlight its main contribution: the inflation of false positives when using several pipelines. Nonetheless, if the editor or R1 would prefer us to include Study 2 in the supplementary materials (possibly with the trimmed means method) we would be happy to do so.

Q11	Step 1: is the proportion of trials per condition?	Yes, in both cases we manipulated the proportion of trials per condition separately. We have clarified it in the ms.
Q12	“The difference in means between the two artificial conditions was calculated and added to the null distribution” - do you mean the value was saved, so at the end you had a distribution of 1,000 differences under the assumption of exchangeability of trials (not participants)?	As study 2 is no longer included, this comment no longer applies.
Q13	“we also included in the null distribution the real observed difference between means corresponding to the original” - alternatively, and perhaps easier to explain, you could simply replace 0 p values with 1/1000.	In fact, it makes a tiny difference: including the observed value in the null has an impact in all p-values, not just on those equal to 0.
Q14	## Discussion The proportion of false positives is only expected to be at 0.05 if the two distributions compared have the same skewness. When distributions differ in skewness, the number of false positives increases. In the presence of outliers, it goes down. So it is misleading to describe an expected 0.05 nominal level when using only one preprocessing pipeline.	As shown in the simulations below, the reviewer is correct, and it occurs so when the distributions are identical. And that is precisely the point of this paper that when distributions are identical the false positive rate increases with the number of applied PPs (flexibility in the analysis). It is not the point of this paper to compare distributions with different degrees of skewness. We have removed the word expected and only reported the false positive rate. We have also added the following paragraph to the discussion, so readers are aware of the difference in skewness as a source of false positives: “We must highlight that flexibility in data preprocessing is not the only factor with an impact on the false-positive rate in analyses of RTs. For example, differences in skewness between conditions can also raise the number of false positives above 5% [14] (please, that no difference in skewness was present in the simulations reported in this study as our interest was to detect false positives when distributions in both conditions were identical). Therefore, researchers must be aware of other possible sources of false positives even when only using a single preregistered analysis pipeline.”
Q15	Using the mean or the median addresses different questions about the data, and both have advantages and limitations in terms of power and bias depending on the shape of the populations. This should be mentioned, so that readers are not under the impression that the mean and the median are interchangeable.	We now address this bias in the mean and median (and its relevance for RT analysis) at the end of the introduction and we refer readers to the appropriate literature dealing with the pros and cons of each preprocessing pipeline at the end of the introduction.

mu1	sigma1	tau1	mu2	sigma2	tau2	fp	n_iter
600	50	0	300	50	300	0.052030	1e+06
600	50	0	400	50	200	0.051929	1e+06
600	50	0	500	50	100	0.051392	1e+06
600	50	0	600	50	0	0.050247	1e+06

Table RI. Number of significant tests when testing a million times, 100 participants in two conditions each with 100 trials using a paired t-test. RTs are simulated using exGaussian distributions, parametrized for the first condition using mu1, sigma1 and tau1, and for the second condition using, mu2, sigma2 and tau2. Skewness in distribution 1 is kept constant while it is varied in distribution 2. Note that the mean of the distributions is the same for all distributions.

mu1	sigma1	tau1	mu2	sigma2	tau2	fp	n_iter
300	50	300	300	50	300	0.04914	1e+05
300	50	500	300	50	500	0.04971	1e+05
300	50	1000	300	50	1000	0.04980	1e+05
300	50	2000	300	50	2000	0.04943	1e+05
400	50	200	400	50	200	0.04981	1e+05
500	50	100	500	50	100	0.04928	1e+05
600	50	0	600	50	0	0.05066	1e+05

Table RII. Number of significant tests when testing a million times, 100 participants in two conditions each with 100 trials using a paired t-test. RTs are simulated using ex-Gauss distributions, parametrized for the first condition using mu1, sigma1 and tau1, and for the second condition using, mu2, sigma2 and tau2. In this case, both conditions come from identical distributions, but we have included larger tau values to assess if longer tails would increase the number of false positives.

References:

- Rousselet, G. A., Pernet, C. R., & Wilcox, R. R. (2019). A practical introduction to bootstrap: a versatile method to make inferences by using data-driven simulations. *PsyArxiv*. <https://doi.org/10.31234/osf.io/h8ft7>
- Rousselet, G. A., & Wilcox, R. R. (2018). Reaction times and other skewed distributions: problems with the mean and the median. *BioRxiv*, 383935. <https://doi.org/10.1101/383935>

Appendix B

Reviewer Comment	Author's Response
The authors have addressed my comments and significantly improved the article. I have only minor comments and suggestions at this stage. I don't think the permutation test results should be included, even as supplementary information, unless the simulations are done at the participant level, as in study 1, and false positives are assessed too. Otherwise some readers might think it is ok to combine trials across participants, a practice that alters degrees of freedom and tests a very different hypothesis than what researchers typically have in mind: https://psyarxiv.com/jqw35 https://elifesciences.org/articles/48175	We agree with the reviewer. We must have made some mistake when uploading files in the last submission, as any mention to the permutation study was removed during the last review, including in the supplementary materials. We apologize for this. We have checked that all mention to the permutation approach has been removed. We would like to note, however, that although the permutation was done at the trial level, the observation unit was still the participant, as trials were shuffled across conditions but within participants.
I still do not understand how many simulation iterations were performed. I tend to think of a simulation iteration as a unit in which participants and trials are sampled, tests are applied and results recorded. I would not describe a simulated observation as a simulation. And I do not think this is typically the case - e.g. https://onlinelibrary.wiley.com/doi/full/10.1002/sim.8086 So in your study, the most important information is the number of values used to compute the proportion of false positives. From your code, it seems that you performed 10,000 simulations. Is that correct?	There was a total of 1 million iterations per dataset. In each of those iterations we created a sample (participants and trials), assigned half of trials within each participant to two conditions, applied all the preprocessing pipelines and statistical tests. Then we calculated the proportion of iterations in which we obtained a significant p-value. Therefore, the steps taken during our simulations coincide with those described by the reviewer. In fact each of the steps described by the reviewer almost matches each of the steps described in the manuscript (p8). We have clarified this in the manuscript with a new paragraph providing an overview of the simulation and the number of simulations run. This paragraph read as follows: “We ran 1,000,000 iterations for each of the real and simulated datasets. In each iteration, we created a random sample of participants, assigned the trials randomly to either of two conditions, applied the previously described PPs and run a t-test for each of those PPs”
On the OSF, I couldn't find a table describing all the files and folders. I suggest that you add a README file	Given the changes made, we have restructured the whole repository. R

explaining the content of the different folders, and how to re-use the code to reproduce the simulations and investigate the results.	code has been transformed into an R package and each function is now documented. We have included a drake pipeline (https://github.com/ropensci/drake) that will help readers re-use the code and reproduce the analysis from the beginning to the end. We have also added a wiki page in the OSF repository explaining where to find the documentation of each function, how to load the package and the location of the drake plan, as well as some cautions and possible problems the users may find. Also, we have removed any mention to the permutation approach in the repository, to avoid confusion.
Sentence two of the abstract needs to be rephrased to avoid repeating "due to" and to improve clarity: "RTs entail some degrees of freedom of their own, due to the several techniques present due to their non-normal skewed distribution and the potential presence of outliers."	We have modified the sentence.
Page 6: "RT modelling may also have researchers' degrees of freedom" - this is misleading because it suggests that a t-tests and ANOVAs are not models.	We have modified the sentence.
Page 9: "a revision of the literature" should be "a survey"?	We have modified the sentence.
Page 13: in the title of figure 2, I would spell out PP for clarity.	We have modified the figure.
Page 14: "due to the RTs distribution" - this needs rephrasing	We have modified the sentence.
Page 15: "please, that..." -> "please note that..."?	We have modified the sentence